Ficus septica plant extracts for treating Dengue virus in vitro

Huang Nan-Chieh 1
Hung Wan-Ting 2
Tsai Wei-Lun 3
Lai Feng-Yi 4
Lin You-Sheng 5
Huang Mei-Shu 5
Chen Jih-Jung 6
Lin Wei-Yu 7
Weng Jing-Ru jrweng@mail.nsysu.edu.tw 8
Chang Tsung-Hsien changth@vghks.gov.tw 5 9
1 Department of Family Medicine, Zuoying Branch of Kaohsiung Armed Forces General Hospital , Kaohsiung , Taiwan
2 Section of critical care medicine, Kaohsiung Veteran General Hospital , Kaohsiung , Taiwan
3 Division of Gastroenterology, Department of Internal Medicine, Kaohsiung Veterans General Hospital , Kaohsiung , Taiwan
4 Deparment of Nursing, Shu-Zen College of Medicine and Management , Kaohsiung , Taiwan
5 Department of Medical Education and Research, Kaohsiung Veterans General Hospital , Kaohsiung , Taiwan
6 Faculty of Pharmacy, National Yang-Ming University , Taipei , Taiwan
7 Department of Pharmacy, Kinmen Hospital , Kinmen , Taiwan
8 Department of Marine Biotechnology and Resources, National Sun Yat-sen University , Kaohsiung , Taiwan
9 Department of Medical Laboratory Science and Biotechnology, Chung Hwa University of Medical Technology , Tainan , Taiwan
Tandon Ravi
Electronic publication date: 2017 Jun 8
Publication date: 2017
Volume: 5
Electronic Location ID: e3448
Received 2016 Dec 23; Accepted 2017 May 19
Copyright: ©2017 Huang et al.
Copyright year: 2017
Copyright holder: Huang et al.
License: This is an open access article distributed under the terms of the Creative Commons Attribution License, which permits unrestricted use, distribution, reproduction and adaptation in any medium and for any purpose provided that it is properly attributed. For attribution, the original author(s), title, publication source (PeerJ) and either DOI or URL of the article must be cited.
License URL: https://creativecommons.org/licenses/by/4.0/

Keywords: Ficus septica, Dengue virus, Aichi virus, Crude extract

Funding: Kaohsiung Armed Forces General Hospital Zuoying Branch ZAFGH105-10 Ministry of Health and Welfare MHW-10347 Kaohsiung Veterans General Hospital, Taiwan VGHKS105-134 This work was supported in part by grants from Kaohsiung Armed Forces General Hospital Zuoying Branch (ZAFGH105-10), the Ministry of Health and Welfare (MHW-10347) and Kaohsiung Veterans General Hospital, Taiwan (VGHKS105-134). There was no additional external funding received for this study. The funders had no role in study design, data collection and analysis, decision to publish, or preparation of the manuscript.

==============================
Dengue virus types 1-4 (DENV-1-4) are positive-strand RNA viruses with an envelope that belongs to the Flaviviridae. DENV infection threatens human health worldwide. However, other than supportive treatments, no specific therapy is available for the infection. In order to discover novel medicine against DENV, we tested 59 crude extracts, without cytotoxicity, from 23 plants in vitro; immunofluorescence assay revealed that the methanol extracts of fruit, heartwood, leaves and stem from Ficus septica Burm. f. had a promising anti-DENV-1 and DENV-2 effect. However, infection with the non-envelope picornavirus, Aichi virus, was not inhibited by treatment with F. septica extracts. F. septica may be a candidate antiviral drug against an enveloped virus such as DENV.

Introduction

Dengue fever is an acute infectious disease caused by dengue virus (DENV), which is transmitted by mosquitoes to humans; about 50 million people are infected per year worldwide (Guzman et al., 2010). According to different serotypes of the virus, four types (DENV types 1-4) are divided. Each type has the ability to cause disease. DENV infection causes varying degrees of disease manifestation, such as self-limited febrile dengue fever, skin rash or drowsiness, agitation, liver enlargement, or dengue hemorrhagic fever (DHF) and even death. A second DENV infection may lead a life-threatening dengue shock syndrome (DSS) (Abel, Liautaud & Cabie, 2012; Kyle & Harris, 2008; Martina, Koraka & Osterhaus, 2009). Currently, no specific therapy is available for the infection other than supportive treatments (Guzman et al., 2010).

The identification and use of medicinal plants for treatment of various diseases has been done throughout human history. Certain medicinal plants also show antiviral activity, such as Carissa edulis Vahl against herpes simplex virus, Geranium sanguineum L. against influenza virus A, Boehmeria nivea L. against hepatitis B virus, Saxifraga melanocentra Engl. & Irmsch. against hepatitis C virus, Lycoris radiata (L’Hér.) Herb. against severe acute respiratory syndrome-associated coronavirus and Phyllanthus amarus Schum. & Thonn. against HIV (Mukhtar et al., 2008). In addition, the neem (Azadirachta indica A. Juss.) showed potential inhibition of DENV-2 replication (Parida et al., 2002). Thus, discovering a novel antiviral medicine from medical plants would be a promising strategy.

In this study, we collected 23 plants from among Taiwanese folk medicinal plants for screening anti-DENV herbs. We also investigated the antiviral effect on Aichi virus (AiV), a pathogenic picornavirus that causes gastroenteritis. Among these plants, we found that F. septica Burm. f. could be a potential medicinal plant against DENV.

Materials and Methods

Virus and cell line

We used local Taiwanese strains of DENV-1 766733A and DENV-2 PL046 (Genbank accession no. AJ968413.1) isolated from patients with dengue fever (Lin et al., 1998). The viruses were propagated in mosquito cell line C6/36 (ATCC: CRL-1660) grown in RPMI 1640 medium containing 5% fetal bovine serum (FBS). The human Aichi virus (AiV) was isolated from a newborn with diarrhea in Taiwan and propagated in Vero cells (ATCC: CCL-81) (Chen et al., 2013). Vero cell, A549 human lung epithelial carcinoma cells (ATCC: CCL-185) and Huh7.5 human hematoma cells (ATCC® PTA-8561™) were cultured in DMEM supplemented with 10% fetal bovine serum (FBS; Thermo Fisher, Waltham, MA, USA). HepG2 human hepatocellular carcinoma cells (ATCC® HB-8065™) and WS1 human fetal skin normal fibroblasts (BCRC: 60300) were cultured in MEM medium supplemented with 10% FBS and non-essential amino acids (NEAA; Gibco, Thermo Fisher, Waltham, MA, USA).

Plant materials

All plants were purchased from a traditional herb shop or a Chinese medicinal herb shop in Taiwan. The plants were identified by one of the co-authors, Dr. Wei-Yu Lin (October, 2008 to May, 2014). Those plants are Alisma orientalis (Sam.) Juz., Asparagus cochinchinensis (Lour.) Merr., Broussonetia papyrifera (L.) L′Herit. ex Vent., Catharanthus roseus (L.) G. Don, Clausena excavate Burm. f., Cinnamomum insulari-montanum Hayata, Cornus officinalis Torr. ex Dur., Euonymus japonicas Thunb., Elaeocarpus sylvestris (Lour.) Poir., Fraxinus griffithii C. B. Clarke, Ficus septica, Ficus sarmentosa B. Ham. ex J. E. Sm. var. henryi (King ex D. Oliver) Corner, Garcinia subelliptica Merr., Lumnitzera racemosa Willd., Litchi chinensis Sonn., Phytolacca americana L., Pueraria lobata (Willd.) Ohwi ssp. thomsonii (Benth.) Ohashi & Tateishi, Sida acuta Burm. f., Sambucus chinensis  Lindl., Scrophularia ningpoensis Hemsl, Saurauia tristyla var. oldhamii (Hemsl.) Finet & Gagnep., Tribulus terrestris L., Xanthium sibiricum Patrin ex Widder and Strophanthus divaricatus (Lour.) Hook. Edt Arn. We included material from the whole plant, root, leaves, stem, fruit, pericarp, root bark, flower or heartwood.

Plant extracts

Materials of plant species were ground, extracted with the indicated solvent for one week. The extracts were concentrated under vacuum. Plant materials of the species no. 13 (leaves) was subjected to an additional one-step extraction with ethyl acetate (EtOAc) and filtered and dried as described before. The volumes (5 ml) of solvents were used per gram of plant material.

Cell proliferation assay

WST-1 assay (Roche, Basel, Switzerland) was used to monitor cell proliferation (Chou et al., 2014); cells were trypsinized and resuspended in culture medium, then plated at 5 × 103 cells per well in 96-well plates and incubated overnight. After plant extracts treatment for 48 h, the cells were incubated with 10μl WST-1 reagent for 2 h. The cell viability was quantified by multi-well spectrophotometry (Anthos, Biochrom, Cambridge, UK). The absorbance at 450 nm was monitored, and the reference wavelength was set at 620 nm.

Treatment

In the extracts screening of DENV inhibition, the cells (5 × 103 cells) were treated with plant extracts with serial dilution dose of 100, 50, 25, 12.5, 6.25, 3.125 or 1.56μg/ml or DMSO solvent control for 3 h. Then, these cells were infected by DENV-1, DENV-2 and AiV infection (multiplicity of infection [MOI]  = 2.5). After 2 h virus adsorption, the medium mixture was replaced by fresh growth medium. At 42 h after infection, the virus-infected cells were analyzed by immunofluorescence assay. In another experiment, the viral stocks of DENV-1, DENV-2 and AiV were preincubated with a series of dilution doses of leaf methanol extract of F. septica (FS-(L)-M) for 1 h at room temperature. The mixture of virus plus the plant extract was then used to infect A549 cells.

Immunofluorescence assay

Immunofluorescence assay was conducted to determine the DENV and AiV infectivity as we previously described (Chen et al., 2013; Wang et al., 2015). In brief, cells were fixed with 4% paraformaldehyde for 30 min, then permeabilized with 0.5% Triton X-100 for 10 min. After two washes with phosphate buffered saline (PBS), cells were blocked with 10% skim milk in PBS. The cells infected with DENV or AiV were detected by antibody against NS3 (Yao-Houng, Biotechnology, Taipei) or anti-AiV VP1 antibody followed by IRDye 800 CM goat anti-mouse or -rabbit IgG (Li-Cor, Lincoln, NE, USA) or Alexa fluor 488 conjugated anti-mouse IgG (Thermo Fisher, Waltham, MA, USA) . The fluorescence intensity was measured and quantified by using the Li-Cor odyssey CLx imaging system or fluorescence microscopy (Zeiss, AX10).

Statistical analysis

Significant differences between groups were analyzed by 2 tailed Student t test with the software GraphPad Prism 6 (La Jolla, CA, USA). Data are presented as mean ± SD. P < 0.05 was considered statistically significant. The statistical datasets are showed in the supplementary information.

Results and Discussion

We aimed to reveal a medicinal plant candidate against DENV. We extracted 70 different crude compounds from materials of 24 plants. DENV-caused respiratory disease was revealed (Rodrigues et al., 2014; Wang et al., 2007), in addition, our previous study showed that the lung carcinoma A549 cells were well susceptible target cells for DENV, these cells have been used in the study model of viral-host interaction. Thus, A549 cells were applied as the screening model in this study (Chang, Liao & Lin, 2006). The cytotoxic effect of the extracts was evaluated in lung carcinoma A549 cells by WST-1 cell proliferation assay. Except S. divaricatus, extracts from other 23 plants revealed no cytotoxicity effect at the maximum tested concentration, 100 μg/ml. Therefore, S. divaricatus extracts were excluded in the antiviral screening (Supplementary information, Table S1).

Table 1 The IC50 of plant crude extracts against Dengue virus type 2 infection.

No.	Botanical name	Part of plant	Extract	Abbreviation of crude extract	IC50 (μg/ml) of viral inhibition	P value (IC50 vs. Ctrl)*	
1	Alisma orientalis (Sam.) Juz.	Whole plant	Methanol	Ao-(WP)-M	>100		
2	Asparagus cochinchinensis (Lour.) Merr.	Root	Methanol	ACM (R)-M	>100		
3	Broussonetia papyrifera (L.) L′Herit. ex Vent.	Leaves	Methanol	BP (L)-M	>100		
4	Catharanthus roseus (L.) G. Don	Whole plant	Methanol	CaR-(WP)-M	>100		
5	Clausena excavata Burm. f.	Leaves	Methanol	Ce-(L)-M	>100		
6	Cinnamomum insulari-montanum Hayata	Leaves	Methanol	CiM-(L)-M	>100		
7	Cornus officinalis Torr. ex Dur.	Whole plant	Acetone	CO-(WP)-A	>100		
Whole plant	Methanol	CO-(WP)-M	>100		
8	Euonymus japonicus Thunb.	Leaves	Acetone	EJa-L-A	>100		
9	Elaeocarpus sylvestris (Lour.) Poir.	Leaves	Acetone	ES-(L)-A	>100		
Leaves	Chloroform	ES-(L)-C	>100		
Leaves	Methanol	ES-(L)-M	>100		
Stem	Methanol	ES-(S)-M	>100		
10	Fraxinus griffithii C. B. Clarke	Leaves	Acetone	FG-(L)-A	>100		
Leaves	Chloroform	FG-(L)-C	>100		
Leaves	Methanol	FG-(L)-M	>100		
11	Ficus septica Burm. f.	Root Bark	Acetone	FS-(RB)-A	3.05 ± 0.75	<0.001	
Leaves	Methanol-Ethyl acetate	FS-(L)-M-ET	24.62 ± 4.04	<0.001	
Fruit	Methanol	FS-(F)-M	37.46 ± 12.3	<0.001	
Heartwood	Methanol	FS-(HW)-M	24.07 ± 13.18	<0.001	
Leaves	Acetone	FS-(L)-A	25.58 ± 9.13	<0.001	
Leaves	Chloroform	FS-(L)-C	>100		
Leaves	Methanol	FS-(L)-M	18.37 ± 10.6	<0.001	
Stem	Methanol	FS-(S)-M	35.64 ± 21.2	<0.001	
12	Ficus sarmentosa B. Ham. ex J. E. Sm. var. henryi (King ex D. Oliver) Corner	Leaves	Acetone	FSVH-(L)-A	72.04 ± 14.5	<0.05	
Leaves	Chloroform	FSVH-(L)-C	>100		
Leaves	Methanol	FSVH-(L)-M	>100		
Stem	Methanol	FSVH-(S)-M	>100		
13	Garcinia subelliptica Merr.	Flower	Methanol	GS-(F)-M	>100		
14	Lumnitzera racemosa Willd.	Leaves	Methanol	Lr-(L)-M	>100		
15	Litchi chinensis Sonn.	Leaves	Acetone	LC-(L)-A	>100		
Leaves	Chloroform	LC-(L)-C	>100		
Leaves	Methanol	LC-(L)-M	>100		
Stem	Acetone	LC-(S)-A	>100		
Stem	Chloorform	LC-(S)-C	>100		
Stem	Methanol	LC-(S)-M	>100		
Fruit	Acetone	LC-(FR)-A	>100		
Pericarp	Acetone	LC-(Peri)-A	>100		
Pericarp	Methanol	LC-(Peri)-M	>100		
16	Phytolacca americana L.	Whole plant	Acetone	PA-(WP)-A	>100		
Whole plant	Chloroform	PA-(WP)-C	>100		
Whole plant	Methanol	PA-(WP)-M	>100		
17	Pueraria lobata (Willd.) Ohwi ssp. thomsonii (Benth.) Ohashi & Tateishi	Whole plant	Methanol	PL -(WP)-M	>100		
18	Sida acuta Burm. f.	Whole plant	Methanol	Sa-(WP)-M	>100		
19	Sambucus chinensis  Lindl.	Whole plant	Acetone	Scl-(WP)-A	>100		
Whole plant	Chloroform	Scl-(WP)-C	>100		
Whole plant	Methanol	Scl-(WP)-M	>100		
20	Scrophularia ningpoensis Hemsl	Whole plant	Methanol	SN-(WP)-M	>100		
21	Saurauia tristyla var. oldhamii (Hemsl.) Finet & Gagnep.	Leaves	Chloroform	STV-(L)-C	>100		
Leaves	Methanol	STV-(L)-M	>100		
Leaves	Acetone	STV-(L)-A	>100		
22	Tribulus terrestris L.	Fruit	Acetone	TT-(Fr)-A	>100		
Fruit	Methanol	TT-(Fr)-M	>100		
Fruit	Chloroform	TT-(Fr)-C	>100		
Fruit	Methanol	TT-(Fr)-M	>100		
Whole plant	Acetone	TT-(WP)-A	>100		
Whole plant	Chloroform	TT-(WP)-C	>100		
Whole plant	Methanol	TT-(WP)-M	>100		
23	Xanthium sibiricum Patrin ex Widder	Fruit	Chloroform	XS-(Fr)-C	>100		
Fruit	Methanol	XS-(Fr)-M	>100		
Notes.

* P < 0.05 estimated by 2-tailed Student t test (IC50 vs. Control).

The immunofluorescence results indicated that the F. septica materials root bark acetone (FS-(RB)-A) and fruit methanol extracts (FS-(Fr)-M), heartwood methanol extract (FS-(HW)-M), leaf acetone and methanol extracts (FS-(L)-A, FS-(L)-M), and stem methanol extract (FS-(S)-M) significantly inhibited DENV-2 infection, with IC50 from 3.05 ± 0.75 to 37.46 ± 12.3μg/ml (Table 1). In addition, leaf extracts of F. sarmentosa var. henryi showed an anti-DENV-2 effect, with IC5072.04 ± 14.5μg/ml, which was higher than for the extracts of F. septica (Table 1).

In addition to DENV-2, the DENV-1 was inhibited by FS-(L)-M in A549 cells (IC50 = 28 ± 10.4μg/ml); however, the AiV infection was not affected by FS-(L)-M treatment (Figs. 1A and 1B). DENV infection-mediated liver disorder was reported (Samanta & Sharma, 2015; Tristao-Sa et al., 2012), therefore, the F. septica against DENV1 and DENV-2 but not AiV were also confirmed in the hematoma cell lines, HepG2 and Huh7.5 cells (Figs. 1C and 1E). Interestingly, FS-(L)-M showed more potent anti-DENV-1 and DENV-2 effect in HepG2 cells (IC50 = 10.1 ± 2.4μg/ml and 12.2 ± 2.1μg/ml, respectively) then in Huh7.5 cells (IC50 = 39.8 ± 6.9μg/ml and 21.9 ± 3.9μg/ml, respectively) (Figs. 1D and 1F). Moreover, skin normal fibroblasts (Wang et al., 2015), WS1 cells were used as the non-cancerous cells for testing the anti-viral effect of FS-(L)-M. The similar results showed that the FS-(L)-M inhibited DENV-1 and DENV-2 in WS1 cells with IC5013.3 ± 2.6μg/ml and 10.6 ± 1.1μg/ml, respectively (Figs. 1G and 1H). A higher dose of F. septica (IC5041.1 ± 6.7μg/ml) against AiV was determined in WS1 cells, which was not showed in other tested cells types. This data implicated a cell type-specific manner of AiV inhibition by F. septica. However, the precise mechanism remains to be further explored.

Figure 1 Ficus septica leaf methanol extract inhibits DENV infection in various cell types.

(A, C, E and G) A549, HepG2, Huh7.5 and WS1 cells (3 × 104) were incubated with various doses of F. septica leaf methanol extract (FS-(L)-M, 3.125 ∼ 50μg/ml) or DMSO solvent control (Ctrl) for 3 h before dengue virus type I and type II (DENV-1, DENV-2) and Aichi virus (AiV) infection at MOI = 2.5. After 2 h of virus adsorption and 42 h incubation, the immunofluorescence assay was performed to detect the viral infected cells. (B, D, F and G) Fluorescence intensity was measured and quantified by the Li-Cor odyssey CLx imaging system. The IC50 of FS-(L)-M on virus inhibition was indicated. The data are mean ± SD (n = 6). * P < 0.05, ** P < 0.01, *** P < 0.001 by two-tailed Student t test.

Figure 2 Leaf methanol extract of Ficus septica inhibits enveloped viral infection.

(A) DENV-1, DENV-2 and AiV viral stocks were incubated with various doses of FS-(L)-M for 1 h at room temperature before used to infect A549 cells (3 × 104) at MOI = 2.5. After 2 h of virus adsorption, the virus–compound mixture medium was replaced by fresh growth medium for further 42 h incubation. DENV- or AiV-infected cells were detected by immunofluorescence assay. (B) Fluorescence intensity was measured and quantified by the Li-Cor odyssey CLx imaging system. The data are mean ± SD from three independent experiments. *** P < 0.001 by two-tailed Student t test.

In order to understand whether F. septica leaf extracted with methanol, FS-(L)-M, directly inactivated DENV, we preincubated viral stocks of DENV-1, DENV-2 and AiV with a series of dilution doses of FS-(L)-M for 1 h at room temperature. After virus adsorption, the mixture of virus plus the plant extract was replaced by fresh growth medium. At 42 h after infection, the virus-infected cells were analyzed by immunofluorescence assay (Fig. 2A). FS-(L)-M significantly inhibited DENV-1 (IC50 = 17.4 ± 4.6μg/ml) and DENV-2 (IC50 = 15.8 ± 2.5μg/ml) but not AiV (Fig. 2B). Thus, FS-(L)-M directly impaired enveloped viral particles but not non-enveloped virus.

In this study, F. septica extracts had a promising anti-DENV-1 and -DENV-2 effect. Nevertheless, the non-enveloped picornavirus AiV was not efficiently inhibited by F. septica extract. Thus, F. septica would be a possible antiviral drug candidate against enveloped virus, such as DENV.

F. septica, a member of the family Moraceae, is widely distributed in the tropic and subtropic regions of the Western Pacific area (Weekly Asahi Encyclopedia, 1995). In Papua New Guinea, this plant has been used as a medicine to treat illnesses such as cold, fever, gastralgia and fungal and bacterial disease (Holdsworth, Hurley & Rayner, 1980). Several bioactive compounds from F. septica identified include phenanthroindolizidine and aminocarophenone- and pyrrolidine-type alkaloids (Damu et al., 2005; Damu et al., 2009; Ueda, Takagi & Shin-ya, 2009). Among them, compounds of ficuseptine, 4,6-bis-(4-methoxyphenyl)-1,2,3-trihydroindolizidinium chloride and antofine isolated from methanolic extracts of F. septica leaves showed strong antibacterial and antifungal activities (Baumgartner et al., 1990). In addition, some alkaloids, including dehydrotylophorine, dehydroantofine and tylophoridicine, isolated from methanolic extracts of F. septica twigs showed antimalarial activity (Kubo et al., 2016).

Here, we reveal a new bioactivity of F. septica against dengue virus. Importantly, an enveloped virus but not non-enveloped virus is sensitive to the extract pretreatment, which suggests that certain compounds of F. septica might disrupt the DENV envelope structure or interfere with DENV contacting cells. Moreover, the anti-DENV effect of F. septica was demonstrated in the lung and liver cell types with clinical relevant.

F. septica may be a promising medical plant against DENV. The F. septica materials root bark acetone and leaf methanol extracts showed the best anti-DENV efficacy, further identification of the antiviral compounds from these two parts of F. septica would be important for drug development.

Supplemental Information

Table S1 The cytotoxicity of plant crude extracts in A549 cells

Click here for additional data file.

Supplemental Information 2 Table 1 raw data

Click here for additional data file.

Supplemental Information 3 Figure 1 raw data

Click here for additional data file.

Supplemental Information 4 Figure 2 raw data

Click here for additional data file.

Additional Information and Declarations

Competing Interests

Author Contributions

Data Availability

The authors declare there are no competing interests.

Nan-Chieh Huang performed the experiments, analyzed the data, contributed reagents/materials/analysis tools, reviewed drafts of the paper.

Wan-Ting Hung and Wei-Lun Tsai performed the experiments, analyzed the data, reviewed drafts of the paper.

Feng-Yi Lai, Jih-Jung Chen and Wei-Yu Lin contributed reagents/materials/analysis tools.

You-Sheng Lin and Mei-Shu Huang performed the experiments, analyzed the data, prepared figures and/or tables.

Jing-Ru Weng and Tsung-Hsien Chang conceived and designed the experiments, analyzed the data, wrote the paper, prepared figures and/or tables, reviewed drafts of the paper.

The following information was supplied regarding data availability:

The raw data for Table 1, Figure 1, and Figure 2 has been supplied as Supplementary Files. The raw datasets are included in the Material and Methods section of the manuscript.

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
