# Peer review of "Ficus septica plant extracts for treating Dengue virus in vitro"

_PeerJ, doi:10.7717/peerj.3448_

## Round 0.1 · original submission · Major Revisions

Dear Tsung-Hsien,

Thank you for choosing Peer J for publishing your research work. Your manuscript was reviewed by three independent reviewers. As you can see, two of them have expressed some concerns and suggested major revisions. Therefore, I request you to kindly address all the issues as suggested by the reviewers before we consider this manuscript for the publication.

Thank you

Ravi Tandon

Reviewer 1 ·

Basic reporting

The paper reports the efficacy of Ficus septica extracts for DENV treatment.The manuscript has some interesting findings which should be of significance following additional validation.

Experimental design

1. Methods should be more descriptive.
like treatment procedures (line 100-105) should be described in more details.They are confusing at places.
2. Abbreviations should be mentioned at first instance. Like FS-(L)-M. line 103.
3. line 125. reveal a new medicine for DENV. Such statement should be avoided.
4. Authors sholud explain in text why was lung carcinoma cell A549 used for antiviral assay.
A normal non cancer cell line is generally better suited.

Validity of the findings

For findings to be of greater significance, following additional data needs to be generated.
The whole antiviral assay has been done in A549 cells. The authors should crosscheck their results in at least 2 other cells lines for confidence in the findings.These cell lines should be preferably non cancerous and represent viral site of interaction in the system.

Reviewer 2 ·

Basic reporting

no comment

Experimental design

good

Validity of the findings

No comment

Additional comments

Huang et al have screened 59 crude extract out of which methanol extract of fruit, leafs, stem & heartwood of Ficus have shown anti DENV-1 and DENV-2 activity. Verocell and human lung epithelial carcinoma cell line were used for study purposes. cell survival/ cell death have been studied using fluorescence cell imaging system. Data suggests that IC50 value of Ficus extract is significantly low i. e. 15 and 17 microgram / ml in comparison to other plants. Study however, is carried out using crude extract of plant but still remain pertinent in realm of DENV infection having no permanent curing agent. I find study robust and relevant to scientific community engaged in searching viable option to curb DENV menace . Hence i would rather recommend this study to be published without further revision.

Reviewer 3 ·

Basic reporting

literature references

Experimental design

methods described with sufficient detail and information to replicate
research questions are not well defined

Validity of the findings

data is robust and statistically sound

Additional comments

Author is advised to address following comments before the paper could be considered for publication". You can ask them for Minor or major revision whatever you prefer.
In the paper everywhere Ficus septic was written as the botanical name of the plant except the table and supplementary tables which should be Ficus septica because Ficus septica is the correct botanical name of the plant, also correct the plant name in the title.
Botanical name should be written with the authority, when first time used, such as Ficus septica Burm.f.
Please check the botanical name of all the pants.
Plant material should be collected from the nature that’s why more authentic result could be obtained.
Results need more explanation with discussion.
Only one plant part should be used for all plants for concluding the result in more appropriate manner.
Literature was well referenced and relevant.
Data were statistically analysed.
There were some corrections:
Line 51 Lycoris radiate should be Lycoris radiata
Line 58 gasterointenilitis should be gastroenteritis
Line 80 Americana should be americana
Line 89 (5 mL) should be (5 ml)
Line 89 materials should be material
Line 120 antu-mouse should be anti-mouse
Queries:
Why used different solvents for different plants and also for different parts of same plant?
Why used different parts of same plant as well as different plants to observe the viral inhibition?

---

## Round 0.2 · Minor Revisions

Dear Tsung-Hsien,

Thank you for submitting the revised manuscript after implementing the changes suggested by the reviewers. As you can see that one of the reviewers has asked for very minor revision after which the manuscript may be accepted for the publication.

Best regards,

Ravi

Reviewer 1 ·

Basic reporting

answered all the comments.

Experimental design

Corrected as suggested.

Validity of the findings

good.

Additional comments

Can be accepted.

Reviewer 3 ·

Basic reporting

Basic reporting of this paper is satisfactory and has interesting findings.The language of the the paper is simple and good.

Experimental design

Methodology of the paper is now satisfactory.
Research questions defined and meaningful.

Validity of the findings

Data are statistically robust.
The revised manuscript explaining the results in more detail, which is satisfactory.

Additional comments

All corrections were incorporated by the authors in the revised manuscript.
The authors satisfactorily answered the queries.
The revised manuscript required two more corrections, which are as follows:
Line 158 and 160 - F. septic should be F. septica.
After these minor corrections the paper should be worthy for publication.
I appreciate this paper because the findings are valuable

---

## Round 0.3 · accepted · Accept

Dear Tsung-Hsien,

Thank you for submitting the revised manuscript after implementing the changes suggested by the reviewers. I am pleased to inform that your manuscript is accepted for the publication.

Best regards,

Ravi Tandon